



# Constraining Light Dependency in Modeled Emissions Through Comparison to Observed BVOC Concentrations in a Southeastern US Forest

Namrata Shanmukh Panji[1], Deborah F. McGlynn[1], Laura E. R. Barry[2], Todd M. Scanlon[2], Manuel T. Lerdau[2], Sally E. Pusede[2], and Gabriel Isaacman-VanWertz[1]

[1]Department of Civil and Environmental Engineering, Virginia Tech, Blacksburg, VA, 24061, USA
[2]Department of Environmental Sciences, University of Virginia, Charlottesville, VA 22904, USA

**Correspondence:** Namrata Shanmukh Panji (namratapanji@vt.edu) and Gabriel Isaacman-VanWertz (ivw@vt.edu)

**Abstract.** Climate change will bring about changes in meteorological and ecological factors that are currently used in global-scale models to calculate biogenic emissions. By comparing long-term datasets of biogenic compounds to modeled emissions, this work seeks to improve understanding of these models and their driving factors. We compare speciated BVOC measurements at the Virginia Forest Research Laboratory located in Fluvanna County, VA, USA for the 2020 year with emissions esti-
mated by MEGANv3.2. The emissions were subjected to oxidation in a 0-D box-model (F0AM v4.3) to generate timeseries of modeled concentrations. We find that default light-dependent fractions (LDFs) in the emissions model do not accurately represent observed temporal variability of regional observations. Some monoterpenes with a default light dependence are better represented using light-independent emissions throughout the year (LDF$_{\alpha\text{-pinene}}$=0, as opposed to 0.6), while others are best represented using a seasonally or temporally dependent light dependence. For example, limonene has the highest correlation
between modeled and measured concentrations using a LDF=0 for January through April and roughly 0.74-0.97 in the summer months, in contrast to the default value of 0.4. The monoterpenes $\beta$-thujene, sabinene, and $\gamma$-terpinene similarly have an LDF that varies throughout the year, with light-dependent behavior in summer, while camphene and $\alpha$-fenchene follow light-independent behavior throughout the year. Simulations of most compounds are consistently underpredicted in the winter months compared to observed concentrations. In contrast, day-to-day variability in the concentrations during summer months are relatively well captured using the coupled emissions-chemistry model constrained by regional concentrations of NO$_X$ and
O$_3$.

## 1   Introduction

Reactive organic gases are released into the atmosphere on the scale of ∼935 Tg C per year (Safieddine et al., 2017) and are a critical area of research due to their role in forming ozone and particulate matter, which in-turn have detrimental effects on human health, climate change, and air quality (Ebi and McGregor, 2008). Roughly 90% of non-methane organic carbon
is emitted as biogenic volatile organic compounds (BVOCs), from natural sources (Guenther et al., 1995) such as the regular metabolic processes of vegetation and microbial material (Goldstein and Galbally, 2007). Subsequent photochemistry of



these compounds results in the formation of secondary pollutants such as tropospheric ozone and secondary organic aerosols (SOA) (Atkinson, 2000; Atkinson and Arey, 2003; Guenther et al., 1995). Exposure to tropospheric ozone and SOA can cause
short-term effects like eye, nose, throat irritation, and respiratory symptoms. Prolonged exposure may lead to severe issues like increased cancer risk, and Central Nervous System (CNS), liver, and kidney damage, particularly in vulnerable populations (Kampa and Castanas, 2008). Apart from their effects on human health, they cause radiative forcing that impacts global temperatures (Myhre et al., 2014).

BVOC emissions are influenced by various environmental factors, including temperature, light, ozone levels, and other
meteorological conditions. Higher temperatures often lead to increased BVOC emissions due to enhanced metabolic activity in plants (Dindorf et al., 2006; Rasmussen and Went, 1965) and conversely, BVOC emissions can alleviate temperature stresses (Holopainen, 2004; Holopainen and Gershenzon, 2010). Light availability plays a crucial role in the regulation of emissions, with higher radiation levels stimulating photosynthesis and subsequent BVOC production (Sanadze, 1969; Lerdau and Gray, 2003; Dindorf et al., 2006; Li and Sharkey, 2013). Ozone levels can also influence BVOC emissions, with elevated ozone
concentrations exacerbating or inhibiting emission rates depending on duration of exposure (Calfapietra et al., 2013; Lu et al., 2019). However, most studies investigate these effects at the leaf-level or tree-level (Yu and Blande, 2021; Chen et al., 2020; Huang et al., 2018; Kivimäenpää et al., 2016; Helmig et al., 2007) and extrapolation to canopy- or ecosystem-scale impacts can be complex. Furthermore, it can be difficult to decouple multiple competing or complementary effects, such as the chemical destruction of emitted BVOCs by increased ozone masking potential increased emissions, or changes in stomatal uptake under
differing conditions (Fiscus et al., 2005; Herbinger et al., 2007) impacting both BVOC and ozone emission and uptake (Sadiq et al., 2017; Zheng et al., 2015).

In addition to the sensitivity of BVOC emissions to meteorological conditions, they also exhibit plant species-specificity and physiology-specificity (Llusia et al., 2008). Hence, BVOCs are emitted at varying rates and composition due to the different plant species available in a forest ecosystem, and factors such as their leaf age, plant health, and seasonality. Previ-
ously, studies have investigated observed temporal BVOC trends on an ecosystem level (Lindwall et al., 2015; Debevec and Sauvage, 2023) and others have modeled global BVOC emissions (Guenther et al., 2006). Global and/or regional BVOC emissions can be estimated using EPA BEIS (Environmental Protection Agency Biogenic Emissions Inventory System available at https://www.epa.gov/air-emissions-modeling/biogenic-emission-inventory-system-beis) and the Model of Emissions of Gases and Aerosols from Nature (MEGANv3.2 available at https://bai.ess.uci.edu/megan/data-and-code/megan32). In previ-
ous comparisons with short-term observations, MEGAN has been shown to be in good agreement for some compounds and some ecosystems, but have discrepancies in other cases (Sindelarova et al., 2014). The model has been shown to overpredict night-time monoterpene emissions due to combining both light- and temperature-dependent effects to calculate monoterpene emissions (Emmerson et al., 2018; Sindelarova et al., 2014). However, the design of MEGAN to couple with different chemical transport mechanisms has allowed regional and global studies of BVOC concentrations (Situ et al., 2013; Emmerson et al.,
2018), secondary organic aerosol formation (Yang et al., 2011), and ozone production (Liu et al., 2018). Seasonal variations in BVOC emissions are sensitive to temperature-dependent factors and leaf area index (Zhang et al., 2021).



Many of the parameters used by default in MEGAN have been estimated as global averages, but variation is possible in these parameters to better reflect local or regional conditions. Of particular interest to this work is the light-dependence of monoterpenes. Emmerson et al. (2018) studied the effect of light-dependence of monoterpene emissions in MEGAN in southeastern Australia where they concluded that disabling the monoterpene light-dependence improved the otherwise underpredicted local monoterpene estimations. Analysis of long-term speciated BVOC concentrations at our research site in a southeastern US forest has shown that some monoterpenes exhibit light-independent behavior, despite light-dependent defaults in the model, while others have seasonally-dependent light-dependence (McGlynn et al., 2023a), which is not a default capability of the model. In this study, we aim to enhance our understanding of BVOC emissions by probing the species-specific temporal variations of light-dependent fractions to better represent the observations at a regional-scale southeastern US forest.

## 2 Methods

### 2.1 Ground-based measurements

*BVOC concentrations*. Mixing ratios of BVOCs were measured at the Virginia Forest Research Laboratory (VFRL) in Palmyra, Virginia (37.9229°N, 78.2739°W) using an automated gas chromatograph with flame ionization detector (GC-FID) system (McGlynn et al., 2023b). The sample is collected 20m above ground, approximately in the middle of the tree canopy, and passed through a sodium thiosulfate-infused quartz filter (Pollmann et al., 2005) to scrub for ozone. It is then collected onto a multi-bed adsorbent trap which is thermally desorbed once every hour for analysis by the GC-FID (McGlynn et al., 2021). Further details about the instrumental setup, its calibration and operation are available in (McGlynn et al., 2021) and McGlynn et al. (2023b). Currently, 2 years of BVOC mixing ratios are available (McGlynn, 2022) but in this study, we focus on measurements made between January 1, 2020 and December 31st, 2020.

*Meteorological data*. Apart from BVOC measurements, this database consists of carbon dioxide mixing ratios (LI-7500; LI-COR, Lincoln, Nebraska), meteorological conditions such as down-welling shortwave radiation (CNR4; Kipp & Zonen, Delft, Netherlands), temperature & relative humidity (HMP45; Vaisala, Helsinki, Finland), pressure (LI-7500; LI-COR, Lincoln, Nebraska), wind speed and wind direction (CSAT3; Campbell Scientific, Logan, Utah). The solar radiation measurements are made at a height of 41 m and the other measurements at a height of 35 m from the ground. Further, ecological information of the surrounding forest such as species composition and abundance is available at Chan et al. (2011).

### 2.2 Emissions model

*Overview of approach*. BVOC emissions are modeled using MEGAN (Model of Emissions of Gases and Aerosols from Nature), a widely recognized mechanistic model that estimates emissions of BVOCs in the atmosphere originating from terrestrial vegetation (Guenther et al., 2012). Leveraging inputs such as plant species, environmental conditions, and meteorological data, the model predicts BVOC emission rates, allowing us to understand their contributions to atmospheric processes. MEGANv3.2 is used in this work. Fundamentally, MEGAN estimates emissions flux ($F_i$) of chemical species i according to Equation (1)



$$F_i = \gamma_i \sum \epsilon_{i,j} \chi_j \tag{1}$$

where $\epsilon_{i,j}$ is the emission factor at standard conditions for vegetation type $j$. $\chi_j$ represents the fractional area of a model
grid cell covered with vegetation. The emission activity factor ($\gamma_i$) accounts for the environmental and phenological conditions
processes controlling emissions such as light ($\gamma_P$), temperature ($\gamma_T$), leaf age ($\gamma_A$), soil moisture ($\gamma_{SM}$), leaf area index (LAI)
and $CO_2$ inhibition ($\gamma_{CO_2}$) as shown in Equation (2).

$$\gamma_i = C_{CE} LAI \gamma_{P,i} \gamma_{T,i} \gamma_{A,i} \gamma_{SM,i} \gamma_{CO_2,i} \tag{2}$$

Emission activity factors describing light and temperature impacts are divided into light-dependent and -independent com-
ponents as

$$\gamma_{P,i} = (1 - LDF_i) + LDF_i \gamma_{P\_LDF} \quad \text{and} \quad \gamma_{T,i} = (1 - LDF_i)\gamma_{T\_LIF,i} + LDF_i \gamma_{T\_LDF,i} \tag{3}$$

In equation 3, $LDF_i$ is the light-dependent fraction for compound $i$, $\gamma_{P\_LDF}$ is the light-dependent activity factor, which
is based on measurements for isoprene (Guenther et al., 2006), and $\gamma_{T\_LIF,i}$ and $\gamma_{T\_LDF,i}$ are the light-independent and
light-dependent fractions of the temperature activity factor for compound $i$ as described by (Guenther et al., 2006).

Once BVOCs are emitted into the atmosphere, they undergo photochemical oxidation to produce secondary products. There-
fore, to be able to compare our observations of BVOC concentrations with those modeled, we convert the BVOC emission rates
estimated by MEGAN to concentrations using a 0-dimensional box-model. MEGANv3.2-derived BVOC emissions are pro-
vided as inputs to the Framework for 0-D Atmospheric Model (F0AM) box model to simulate photochemistry and estimate
time resolved BVOC concentrations. The framework of this modeling process is shown in Figure 1.

*MEGANv3.2 parameterization*. MEGANv3.2 was run from January 2, 2020 to December 29, 2020. A lambert conformal
projection of the United States was used as the grid area for the simulations, where emissions were modeled in the cells
encompassing the VFRL site. Emission factors ($\epsilon_{i,j}$) for the VFRL site were calculated using the MEGAN Emission Factor
Preprocessor (MEGAN3.21-EFP), available in Python (https://bai.ess.uci.edu/megan/data-and-code/megan32), using the tree
species composition shown in Table S2 based on a previously reported vegetation survey (Chan et al., 2011); canopy type is
26.88% needleleaf (predominantly pine) and 73.12% broadleaf (predominantly oak). The standard conditions emission factor
for each compound and compound class used in this study have been listed in Table S1. Leaf Area Index (LAI) was provided
as an input from the Terra MODIS (Moderate Resolution Imaging Spectroradiometer) data product (Myneni et al., 2015),
extracted as 8-day averages at the coordinates of the VFRL using the Application for Extracting and Exploring Analysis Ready
Samples (A$\rho\rho$EEARS, (AppEEARS Team, 2020)).

Essential meteorological data required by MEGANv3.2 includes temperature, pressure, wind speed, water vapor mixing
ratio, and photosynthetically active radiation (PAR). Measurements of relative humidity (%) were converted to water vapor
mixing ratio (kg/kg) based on temperature and pressure. Down-welling shortwave radiation (W/m$^2$) measurements were mul-
tiplied by 0.5 to convert them to PAR (W/m$^2$) based on previously reported estimates that half of all incoming radiation is



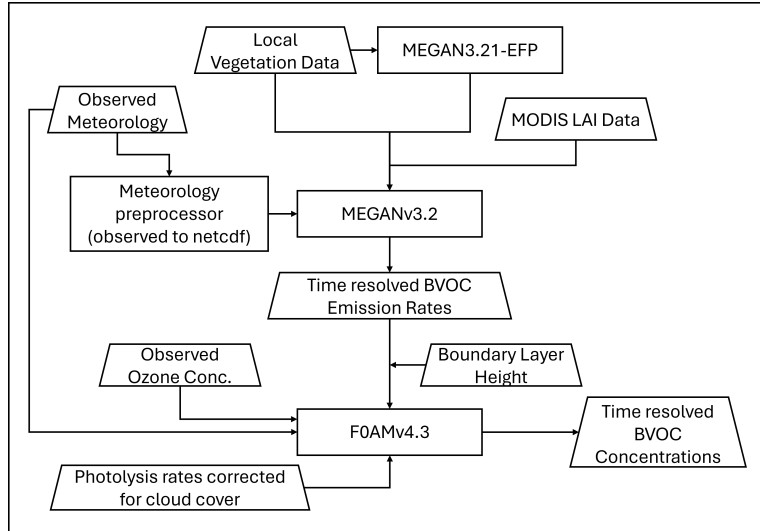

**Figure 1.** Modeling framework used to estimate time resolved local BVOC concentrations at the Virginia Forest Research Laboratory.

photosynthetically active (McCree, 1981). All meteorological measurements at VFRL were resampled to hourly data and con-
verted to a gridded netcdf format suitable as input to MEGANv3.2. Gaps in observed meteorological data were filled with
measurements from two personal weather stations (Weather Underground identifiers KVATROY19 and KVAPALMY31) lo-
cated within 2 miles of VFRL as they are highly correlated with observed VFRL data (see Figure S3 for more information).

A soil type of silty-loam was used for the simulations (Soil Survey Staff, Natural Resources Conservation Service, United
States Department of Agriculture, 2022). Soil temperature and soil moisture were obtained from the ERA5 (the fifth generation
of the European Centre for Medium-Range Weather Forecasts Reanalysis) dataset (Hersbach et al., 2023) using the Open-
Meteo API (Zippenfenig, 2023).

Simulations were run at different light dependent fractions (LDFs) to study the effect of changing emission profiles on
diurnal variability of isoprene, $\alpha$-pinene, limonene, and other monoterpenes. A summary of the model cases are listed in Table
1. The LDFs for $\alpha$-pinene and limonene in MEGANv3.2 were changed to the results of positive matrix factorization (PMF)
analysis conducted on the annual (September 2019 to September 2020) and summer (June, July, and August 2020) BVOC
mixing ratios observed at VFRL (McGlynn et al., 2023a). The resulting timeseries of emission rates (in moles/s) for 200
compounds from MEGANv3.2 were converted to local emission fluxes (in nmol/m$^2$.s) by dividing emissions by the grid cell
area.

## 2.3  F0AM setup

Photochemistry is modeled using the Framework for 0-D Atmospheric Model (F0AM v4.3 (Wolfe et al., 2016)) 0-dimensional
box model that incorporates the Master Chemical Mechanism (MCM v3.3.1, (MCM)), with a starting point of the sample code
'ExampleSetup_DielCycle.m' available with that model. As the VFRL is located in a forest and the closest EPA measurements



**Table 1.** Summary of conditions tested in this study.

| Label | Description | $LDF_{\alpha\text{-pinene}}$ | $LDF_{limonene}$ |
|---|---|---|---|
| Default | LDFs used by default in MEGANv3.2 | 0.6 | 0.4 |
| $PMF_{Annual}$ | Fraction of concentrations attributed to light dependent factor from a PMF analysis of the annual VFRL dataset[†] | 0.02 | 0.57 |
| $PMF_{Summer}$ | Fraction of concentrations attributed to light dependent factor from a PMF analysis of the VFRL summer dataset[†] | 0.03 | 0.67 |
| $Case_{0.0\text{-}1.0}$ | Cases used for correlation studies 1 through 6 | 0, 0.2, 0.4, 0.6, 0.8, 1 | 0, 0.2, 0.4, 0.6, 0.8, 1 |
| Adjusted | Time-dependent LDF based on correlation study | $f_1(t)$ | $f_2(t)$ |

[†] McGlynn et al. (2023a); $f_1(t)$ & $f_2(t)$ as described in Figure 6

of $NO_X$ mixing ratios are close to urban cities, $NO_2$ and NO values were constrained to those measured during the 2013 Southern Oxidant and Aerosol Study (SOAS, (Southern Oxidant and Aerosol Study, 2013)). Because direct measurements of

$NO_X$ at the tower were not available, ozone concentrations were constrained by observations rather than produced through photochemistry in the model. Hourly ozone mixing ratios were provided as those obtained from the nearest Environmental Protection Agency (EPA) AirData Air Quality monitors located at Albermale High School (roughly 15 miles Northwest of VFRL) using the Air Quality System (AQS) API v2 (US Environmental Protection Agency). Since ozone data at this site was only available between March and November of 2020, missing values in January through March were filled with those

during 2021 from Shenandoah National Park (roughly 41 miles North of VFRL) and Prince Edward (roughly 53 miles South of VFRL). Missing values in November and December were filled with ozone mixing ratios measured at VFRL during 2019. Further, $H_2$ and $CH_4$ mixing ratios were constrained to 550 ppb and 1770 ppb respectively. Photolysis rates were calculated by the National Center for Atmospheric Research tropospheric ultraviolet and visible (TUV) transfer model (available at https://www2.acom.ucar.edu/) lookup tables as described by Wolfe et al. (2016). As this model estimates photolysis rates at

clear cloud conditions, the rates were corrected using the method described by Equation 4 (Lu et al., 2017).

$$J_{corrected} = J_{TUV} \times \frac{SWR_{cloud}}{SWR_{clear}} \tag{4}$$

where $J_{corrected}$ is the corrected photolysis rate, $J_{TUV}$ is the photolysis rate calculated from the TUV model lookup tables, $SWR_{cloud}$ is the observed downwelling shortwave radiation at time t in W/m$^2$, and $SWR_{clear}$ is the 7-day rolling maximum of the observed downwelling shortwave radiation in W/m$^2$ to represent clear-sky conditions. A constant 1$^{st}$-order dilution rate of

1/day was used. Observed meteorological data for temperature, pressure, and relative humidity were used. OH concentrations were formed in-situ based on photolysis rates and observed concentrations of ozone and water vapor.

To incorporate the emission flux (in nmol/m$^2$.s) estimated by MEGANv3.2 into the F0AM chemical box model, 0$^{th}$-order reactions were added to the chemical mechanism for each chemical compound, assuming instantaneous emission and mixing



into a box whose size was based on the boundary layer height (obtained from Dan Li (2020)). Observations in the AMDAR
dataset are available regularly but non-continuously throughout the year, and estimation of boundary layer height requires non-
trivial data analyses. Consequently, real-time estimates are not available and a 7-day rolling average of previously published
boundary layer heights for 2007 to 2019 are used (Dan Li, 2020). An average boundary layer height at airports IAD and RDU
are used. Though real-time boundary layer heights for 2020 are reported in the ERA5 dataset, prior work has shown these
estimates to be significantly different than observations, so are not used here (Zhang et al., 2020). The emission rate for F0AM
was hence calculated as in Equation 5,

$$k_{Emission,i}\left(\frac{molecules}{cm^3.s}\right) = F_i\left(\frac{nmol}{m^2.s}\right) \times 6.023 \times 10^{23}\left(\frac{molecules}{mol}\right) \times 10^{-9}\left(\frac{mol}{nmol}\right) \times \frac{1}{BLH(m)} \times 10^{-6}\left(\frac{m^3}{cm^3}\right) \quad (5)$$

where $F_i$ is the emission flux estimated by MEGANv3.2 for compound $i$, and $BLH$ is the boundary layer height in m. These
settings were kept unchanged for all cases listed in Table 1. Results of the F0AM simulations allow us to compare simulated
and observed concentrations as reported in the following section.

## 3   Results and Discussions

### 3.1   Effect of changing LDF on emissions

Emissions peaks during the day, even with relatively high fractions of light-dependent emissions (Figure 2). Overall, light-
dependence has a stronger impact on night-time than on day-time concentrations. By decreasing the LDF of $\alpha$-pinene (Figure
2c) from the default value of 0.6 to nearly 0 (0.02 and 0.03 in Annual and Summer cases respectively), nighttime emissions
increase by 139%-143%, and the daytime emissions increase by 47%-83%. Similarly, by increasing the LDF of limonene
(Figure 2b) from the default of 0.4 to 0.57 and 0.67 in the Annual and Summer cases respectively, we see a decrease in the
nighttime emissions ranging from 28%-45% and daytime emissions by 11%-29%. Since the LDF of isoprene was not changed
over any of the studies, the estimated emission diurnals remain unchanged (overlapping blue, red, and green lines in Figure 2),
but demonstrates the diurnal profile of a fully light-dependent compound (LDF of 1.0). These effects are observed across all
months, though the emission rates in the winter months are approximately 1-2% those in the summer months. Our observations
agree with Emmerson et al. (2018) who note that the nighttime monoterpene emissions in summer increase by 90%-100% when
LDF is turned off.

Given the impact of LDF on night-time emissions coupled with the typically low-boundary layer height at nighttime (which
will present as increased concentrations due to accumulation of emissions), it is important to more closely examine the LDFs
that best represent the observed concentrations of various compounds.



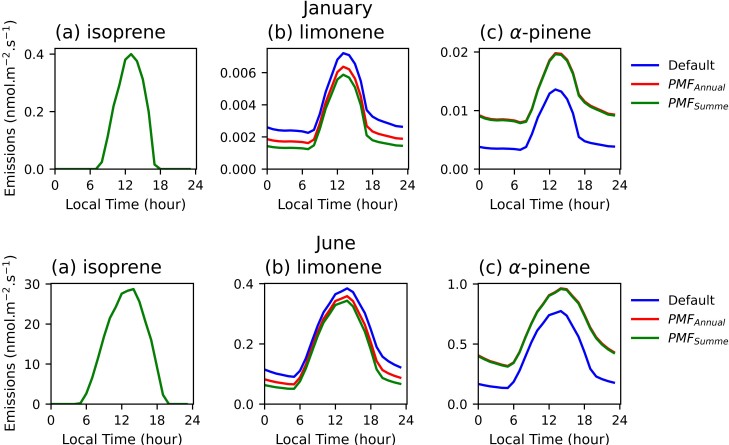

**Figure 2.** Monthly diurnal averages of the (a) isoprene, (b) limonene, and (c) $\alpha$-pinene emissions (in nmol/m$^2$.s) estimated by MEGANv3.2 at VFRL during January (top) and June (bottom) of 2020. The blue, red, and green lines indicate the Default (LDF$_{isoprene}$=1, LDF$_{limonene}$=0.4, LDF$_{apinene}$=0.6), PMF$_{Annual}$ (LDF$_{isoprene}$=1, LDF$_{limonene}$=0.57, LDF$_{apinene}$=0.03), and PMF$_{Summer}$ (LDF$_{isoprene}$=1, LDF$_{limonene}$=0.67, LDF$_{apinene}$=0.02) conditions as described in Table 1.

## 3.2 Measured vs. modeled BVOC concentrations

Modeled concentrations of monoterpenes generally have maxima in the evening, which agrees with $\alpha$-pinene observations but is in contrast to summertime limonene (Figure 3). For all monoterpenes, even though observed wintertime concentrations are lower than those in the summer, the model still substantially underestimates concentrations in winter and spring (January to
May). Prior studies comparing observed concentrations with those simulated were carried out in peak summer and hence find relatively good agreement (Sindelarova et al., 2014). Concentrations of isoprene were below levels of detection of the GC-FID system through the winter, in contrast to low, but non-zero, emissions modeled for isoprene. Modeled and observed concentrations in the summer months are comparable in magnitude for all three compounds in the summer, and some day-to-day variability is also captured (4). Some day-to-day deviations between modeled and measured concentrations are expected, as
boundary layer heights and some atmospheric composition data are averages, which cannot capture real-world variability. Although there are no isoprene emissions during night-time (Figure 3), non-zero night-time isoprene concentrations are observed, which may suggest that nighttime chemistry and/or dilution may not be fully captured (Figure 4).

As noted in (McGlynn et al., 2023a), limonene in summer follow daytime peaks in concentration consistent with partially light-dependent behavior, and night-time peaks during winter. The model fails to capture these observed trends, either with
default LDF values or with those estimated using the fraction of limonene estimated as light-dependent based on factorization ($\sim$60%). In contrast, $\alpha$-pinene has nighttime peaks, which are approximately captured by default values and by changing the LDF to 0.03. Importantly, it is clear that the LDF that best captures observed variability may vary throughout the year, indicating important seasonality.



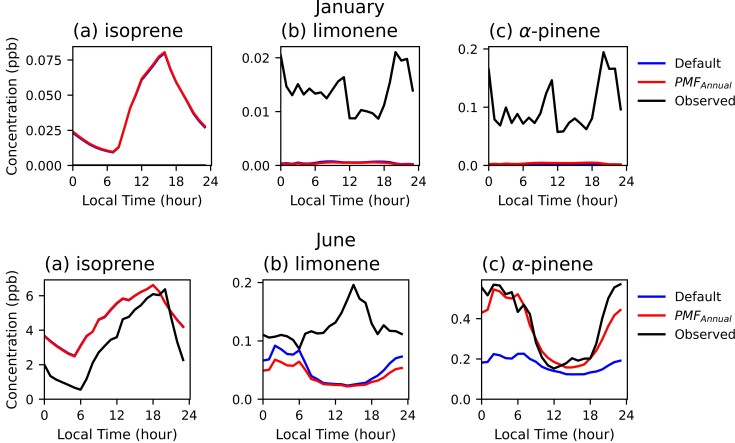

**Figure 3.** Monthly diurnal averages of the (a) isoprene, (b) limonene, and (c) $\alpha$-pinene concentrations (in ppb) estimated by MEGANv3.2 and F0AM at VFRL during January (top) and June (bottom) of 2020. The blue and red lines indicate the Default ($\text{LDF}_{\text{isoprene}}=1$, $\text{LDF}_{\text{limonene}}=0.4$, $\text{LDF}_{\text{apinene}}=0.6$) and $\text{PMF}_{\text{Annual}}$ ($\text{LDF}_{\text{isoprene}}=1$, $\text{LDF}_{\text{limonene}}=0.57$, $\text{LDF}_{\text{apinene}}=0.03$) conditions as described in Table 1, and black lines indicate observed data. Observed concentrations for isoprene were below GC-FID limits of detection for January 2020.

## 3.3 Monthly variation of LDF

To understand the variation of LDF throughout the year, we ran MEGAN simulations at 6 different values of $\text{LDF}_{\alpha\text{-pinene}}$ and $\text{LDF}_{\text{limonene}}$: 0, 0.2, 0.4, 0.6, 0.8, and 1. Emissions from each scenario were provided to the F0AM model and a snapshot of the results from July 2020 are as shown in Figure 5 compared to observed data. As expected from previous studies, concentrations of $\alpha$-pinene correlate well with low values of LDF (5b). While low LDF values for limonene better match observations in absolute magnitude, they substantially invert the observed diurnality. Instead, observed concentrations of limonene correlate

best with high values of LDF (cyan line in Figure 5a) though absolute concentrations are underpredicted. To better quantify the monthly variation of LDF with the highest correlation with observed concentrations, linear interpolation between emissions scenarios was used to estimate emissions at resolution of 0.01 LDF. The light dependency that best correlates with observations varies throughout the year (Figure 6), with a peak in light dependence during the summer and less light-dependence during the rest of the year. Conversely, a constant $\text{LDF}_{\alpha\text{-pinene}}$ of 0 (i.e., light-independent) throughout the year slightly improves the

correlation coefficient.

Simulations were re-run using the LDF value for each month that maximizes correlation (Figure 7, for $\alpha$-pinene see Figure S4). In the month of January, there is no discernible increase in correlation by changing the LDF and the modeled concentrations are extremely low. In the summer (Figure 7a), the correlation between the simulations and the observed concentrations improves from -0.01 to 0.49 by changing the LDF from a default value of 0.4 to a value of 0.97. However, the magnitude

of the modeled concentrations remains low, suggesting underpredicted emission rates. In September (Figure 7b), there is no





**Figure 4.** A snapshot of concentrations (in ppb) of (a) isoprene, (b) limonene, and (c) $\alpha$-pinene estimated by MEGANv3.2 and F0AM at VFRL during January (top) and June (bottom) of 2020. The blue and red lines indicate the Default and PMF$_{Annual}$ conditions as described in Table 1, and black lines indicate observed data. Observed concentrations for isoprene were below GC-FID limits of detection for January 2020.





significant change in the correlation by changing LDF as the diurnal shapes are similar below an LDF of 0.8 but the magnitude of the concentrations at the adjusted LDFs are closer to those observed.

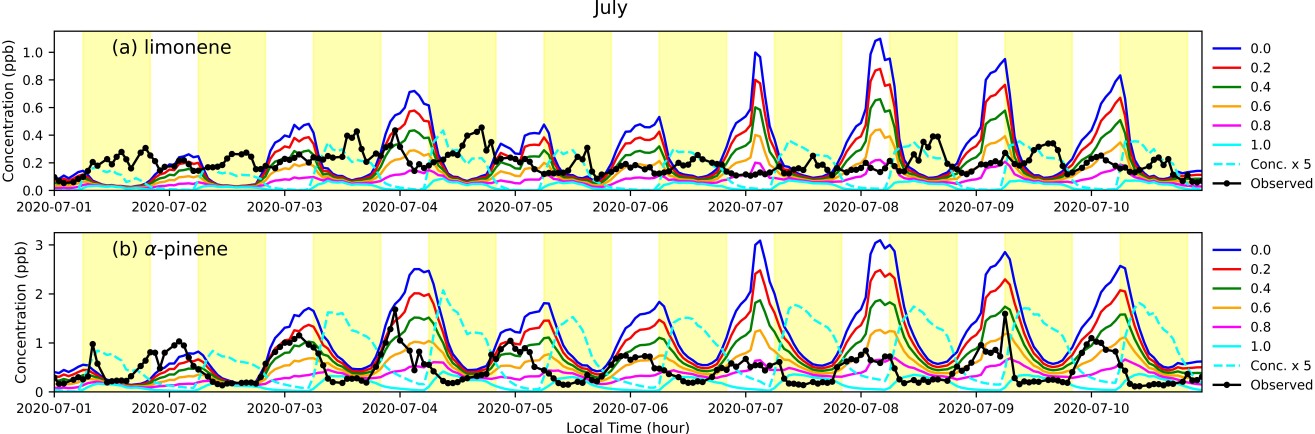

**Figure 5.** A snapshot of (a) limonene and (b) $\alpha$-pinene concentrations (in ppb) estimated by MEGANv3.2 and F0AM at VFRL during and July of 2020. The blue, red, green, orange, magenta, cyan lines refer to simulations with LDF set to 0, 0.2, 0.4, 0.6, 0.8, and 1 respectively (Table 1), and black lines indicate observed data. The dashed cyan line represents the simulated concentrations for LDF=1.0 multiplied by 5. Daytime hours are highlighted in yellow.

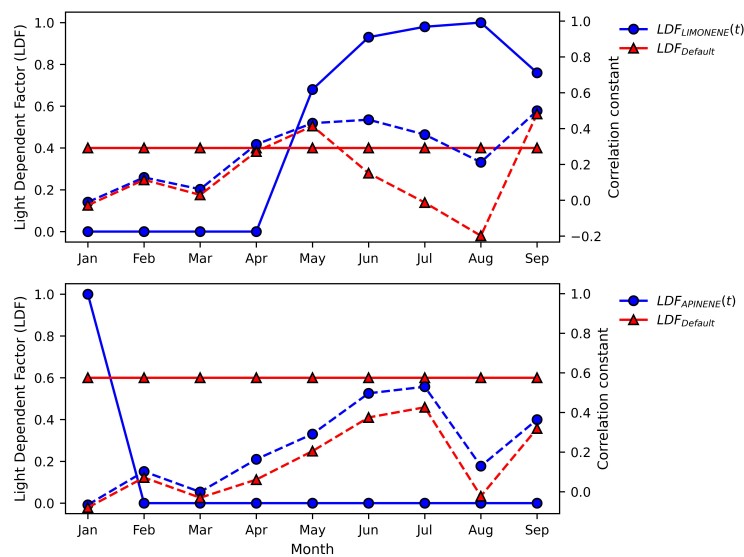

**Figure 6.** Monthly variation of LDF for limonene and $\alpha$-pinene estimated by maximizing the correlation between observed concentrations and those estimated by MEGANv3.2 and F0AM at VFRL. The blue circles and red triangles marked solid lines represent the time-dependent and Default LDF values respectively and the dashed lines represent the corresponding Pearson correlation coefficients.



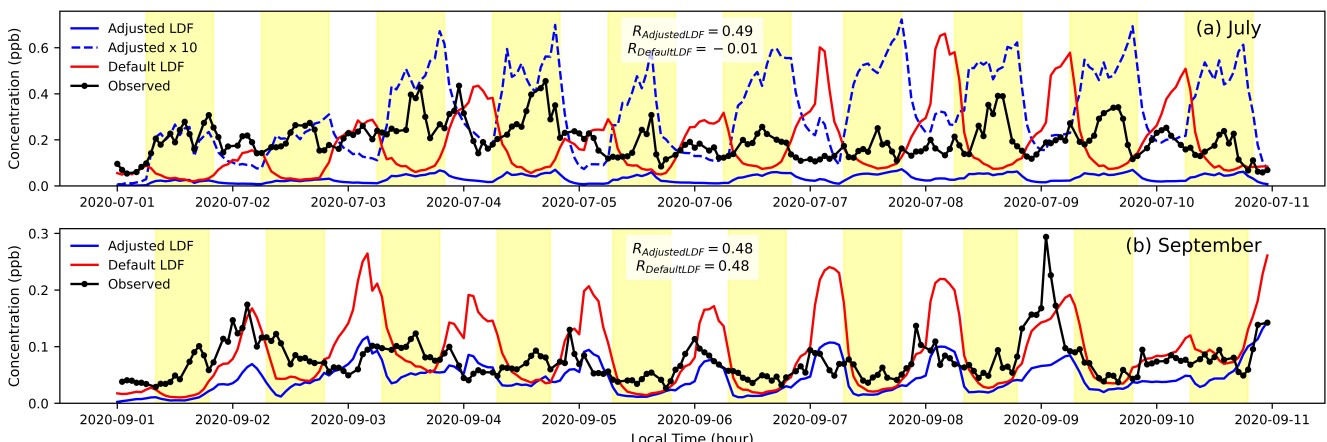

**Figure 7.** A snapshot of limonene concentrations (in ppb) for July, and September of 2020 using the monthly LDF as shown in Figure 6. The dashed blue line represents the simulated concentrations for Adjusted LDF multiplied by 10. The Pearson correlation coefficient values of the adjusted and default modeled (refer to Table 1) concentrations against the observed concentrations are reported as $R_{AdjustedLDF}$ and $R_{DefaultLDF}$.

## 3.4 Extension of F0AM for other BVOCs

Currently, MCM3.3.1 used during F0AM simulations contains only isoprene and 3 monoterpenes: isoprene, $\alpha$-pinene, $\beta$-pinene, and limonene. To simulate the concentrations of other compounds measured at VFRL, we added dummy chemical reactions in the MCM to simulate the chemical loss of other BVOCs emitted into the box for which observational data are available. The reaction rates used are as listed in Table S5.

Following the method described in Section 3.3, the monthly variation of sabinene is as shown in Figure 8 and those of $\alpha$-fenchene, $\beta$-phellandrene, $\beta$-thujene, camphene, tricyclene and $\gamma$-terpinene in Figure S6. Sabinene is observed to be almost completely light dependent in the summer, with strong daytime peaks (Figures 8 and 9). Similarly, variation of LDFs for $\beta$-thujene and $\gamma$-terpinene follows a similar pattern to that of limonene where the LDF peaks in the summer and falls off during the rest of the year. $\beta$-phellandrene shows no significant increase in correlation for LDFs between 0-0.8 but the correlation deteriorates to negative values (Figure S7) if we assume it is completely light-dependent (LDF=1). By default, MEGANv3.2 assigns the limonene compound class LDF for $\beta$-phellandrene and $\gamma$-terpinene (LDF=0.4), and carene compound class LDF for sabinine, $\alpha$-fenchene, $\beta$-thujene, camphene, and tricyclene (LDF=0.2). It is interesting to note that MEGANv3.2 assumes that sabinene, $\beta$-thujene, and $\gamma$-terpinene are partially light-dependent throughout the year, despite more appropriate summertime LDFs of around 1. The results in this study corroborate the PMF observations by McGlynn et al. (2023a) which note that $\beta$-thujene, sabinine, $\gamma$-terpinene, and $\beta$-phellandrene display light-dependent behaviors. Further, $\alpha$-fenchene and camphene behave similar to $\alpha$-pinene where they remain completely light-independent throughout the year (as opposed to the 0.2 LDF assumed in the MEGAN model).



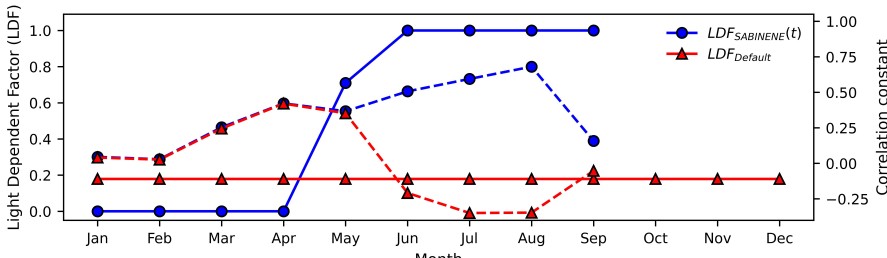

**Figure 8.** Monthly variation of LDF for sabinene estimated by maximizing the correlation between observed concentrations and those estimated by MEGANv3.2 and F0AM at VFRL. The blue circles represent the adjusted LDF values, the red triangles represents the LDF used by default in MEGANv3.2. The blue and red dashed lines present the LDF values and the dashed lines represent the corresponding Pearson correlation coefficients.

Simulations were run at an LDF of 1 (highest correlation, Figure 8) for sabinine in July and September. Upon correcting the LDF to be completely light-dependent, sabinine concentrations peak during the daytime to match the diurnals of observed concentrations. Similar to limonene, the magnitudes remain underpredicated. Overall, the variability of LDF throughout the year and the deviation from the values currently used suggests that there is an important seasonality to LDF that needs to be incorporated into emissions models.

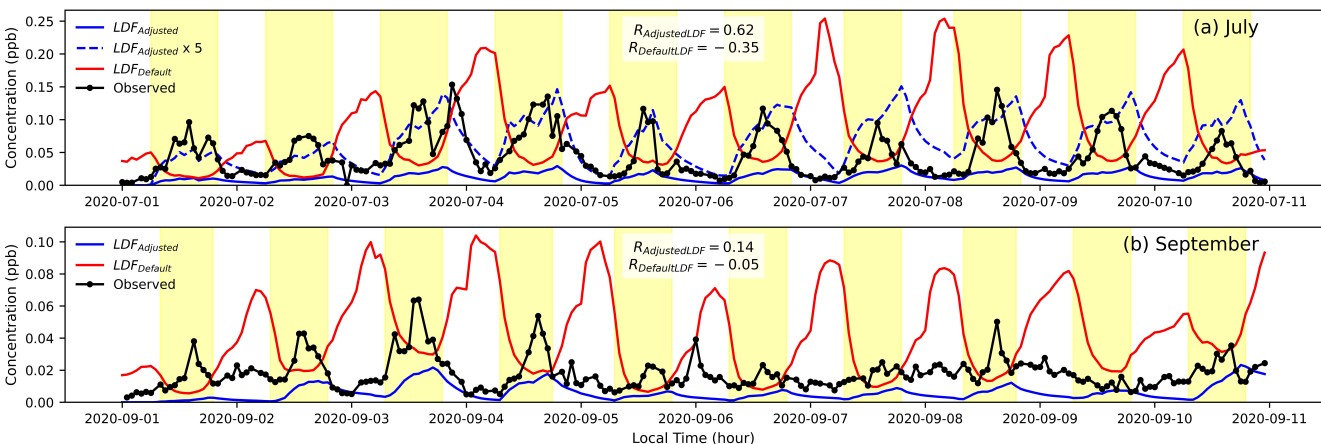

**Figure 9.** A snapshot of sabinene concentrations (in ppb) for July, and September of 2020 using the monthly LDF as shown in Figure 8. The dashed blue line represents the simulated concentrations for adjusted LDF multiplied by 5.

## 4   Conclusions

In this study, we used MEGANv3.2 to simulate BVOC emissions during the year 2020 at a southeastern U.S. forest using local ecological and meteorological data. The photochemistry was simulated by F0AM v4.3 to obtain speciated concentration





timeseries. Prior work at this site shows that the LDFs used in these models contradict the light dependent contribution of
some monoterpenes estimated from observed diurnals (McGlynn et al., 2023a). In this work, we demonstrate that LDFs for
monoterpenes used by default in the models are not consistent with the observed diurnal and seasonal patterns. We observe
an LDF$_{\alpha\text{-pinene}}$ value of 0 (as opposed to 0.6) and a time-dependent LDF$_{\text{limonene}}$ of 0 for January through April and roughly
0.74-0.97 in the summer months as described in Figure 6. Further, we were able to extend the model to simulate concentrations
of other monoterpenes to get speciated information at our local site. We note that LDFs of $\alpha$-pinene-like bicyclic monoterpenes
camphene and $\alpha$-fenchene follow light-independent behavior throughout the year much like the $\alpha$-pinene. Further, molecules
with structures similar to limonene-like such as $\beta$-thujene, and bicylic monoterpenes with a 5-carbon ring such as sabinene,
and $\gamma$-terpinene have an LDF that varies throughout the year (like limonene), with light-dependent behavior in summer and
light-independent behavior in winter and spring. Lastly, we note that the simulations fail to capture observed concentrations
in the winter months where they are consistently underpredicted. In contrast, we are able to capture the day-to-day variability
in the concentrations during summer months using this relatively simple setup of MEGAN and a 0-D box model with 7-day
rolling average of the boundary layer height conditions, and NO$_X$ and O$_3$ concentrations from non-local sources.

This study underscores the need for significant improvements in the LDF within BVOC emission models, particularly for
monoterpenes, to more accurately reflect the complex nature of BVOC emissions under varying lighting conditions. The
observed seasonality in LDF for specific monoterpenes highlights the importance of incorporating temporal changes into
these models. The implications of this research are wide-ranging. Enhancing the precision of BVOC emission models can
lead to more accurate forecasts of atmospheric chemistry, which in turn impacts air quality and climate. Improved models will
also aid in better understanding the role of BVOCs in the formation of ozone and secondary organic aerosols (SOAs), both of
which have significant environmental and health impacts. Additionally, acknowledging the seasonality in LDF can guide the
development of dynamic models that adjust for seasonal variations in monoterpene emissions, thereby increasing the accuracy
of emission inventories, especially in regions with pronounced seasonal shifts. Further, the LDF seasonality has important
implications on the biology underlying terpene production and emission. Earlier studies have suggested a positive relationship
between light dependency and lack of storage of terpene compounds e.g., Lerdau and Gray (2003), and the results presented
here suggest that metabolic pathways and storage processes may vary both among and within compounds over the course of
the year. Compounds showing higher LDFs may be serving as responses to factors that vary with light intensity, so storage is
less beneficial, while those with low LDFs are more analogous to constitutive compounds that function in response to factors
that act independently of light. Similar results have been shown for certain floral volatiles (Theis et al., 2007), but this study is
the first to report such findings for leaf volatiles. Overall, our findings highlight the urgent need for continuous refinement of
BVOC emission models. By incorporating more precise LDF parameters and considering seasonal dynamics, we can deepen
our comprehension of the interactions between the biosphere and atmosphere, and their broader effects on climate and public
health.



*Code and data availability.* Data required to run simulations are provided in Tables plotted in the main results are provided in Tables S1, S2, and S5. Raw data (i.e., meteorological and BVOC concentrations) are available by request by contacting the corresponding author.

*Author contributions.* NSP: conceptualization, data curation, methodology, investigation, writing (original draft), formal analysis, visualization. GI-VW: conceptualization, methodology, writing (review and editing), supervision, funding acquisition. DFM: conceptualization, data
curation, writing (review and editing). LEB: writing (review and editing), data curation. TMD: data curation, writing (review and editing). MTL: conceptualization, writing (review and editing). SEP: conceptualization, writing (review and editing), supervision, funding acquisition.

*Competing interests.* The authors declare that they have no conflict of interest.

*Acknowledgements.* This research was funded by the National Science Foundation (AGS 1837882, AGS 1837891, and AGS 2046367).



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
