# Peer review of "Constraining Light Dependency in Modeled Emissions Through Comparison to Observed BVOC Concentrations in a Southeastern US Forest"

_EGUsphere, 2024_

## Referee Comment (RC2)

Review of the manuscript

**Constraining Light Dependency in Modeled Emissions Through Comparison to Observed BVOC Concentrations in a Southeastern US Forest**

*by Panji et al.*

The submitted manuscript from Panji et al. presents comparison of modeled and measured concentrations of BVOC species, with special focus on monoterpenes, at the Virginia Forest Research Laboratory site (VFRL). The study focuses on the dependence of BVOC emissions on light and therefore proper setting of the light dependence factor (LDF) of individual BVOC compounds in the emission model. To be able to compare the measured concentrations with modeled values, the 0-dimension box model was applied. The authors compare performance of the emission model (MEGANv3.2) through the box model concentrations with MEGAN default LDF values, LDFs obtained from measurements at the VFRL site and LDF values best correlating with the observed concentrations. The paper investigates annual as well as time-varying (monthly) LDF values. The paper suggests new LDF values for selected monoterpene species.

The paper is well written and comprehensively structured. It falls well within the scope of ACP. I suggest accepting the paper for publication after addressing the following questions and minor comments.

Missing measurements of NOx and O3 at the VFRL site at the time of BVOC sampling were substituted by either measurement from previous years or by measurements from 15-53 miles distant stations. Though I understand the need to deal with lack of data, I think these assumptions need at least more discussion of the impact on results. The ozone stations are relatively far away from VFRL with cities such as Charlottesville or Richmond close by that can impact the O3 levels. Do the authors have some evidence that NOx and O3 levels do not have much inter-annual variability at these locations? The NOx and O3 data are crucial for calculation of BVOC concentrations from emissions, therefore can have a substantial impact on the final model to observation comparison.

Apart from LDF values and their seasonal changes, there are other parameters of the emission model that can (partially) explain the discrepancy of the modeled and measured results throughout the year. E.g. emission factors. Though not often used that way, EF can also very during the annual cycle (*Helmig et al., 2013,* https://doi.org/10.1016/j.chemosphere.2013.04.058). The EF intra-annual changes could be another factor that explains a different BVOC concentrations during winter and summer months. Could the authors include this in the discussion?

Can the authors please share their opinion (and include it in the paper 'Discussion' or 'Conclusion' section) on why the modeled concentrations are "consistently underpredicted in the winter months"? Actually, the modeled concentrations of limonene and sabinene are underpredicted also in July (Figs. 7 and 9).

The authors point out well that the LDF values play an important role in the BVOC models and their precise setting is important in order to obtain sensible emission results. Can the authors please elaborate if the LDF values obtained from the measurements at VFRL can be upscaled from this local site to global representation? If the authors think their values could be used for other studies as well, it would be extremely useful if they could add a Table summarising LDF values per species and per month that they recommend to use according to their study. This would be a very good benefit for other emission modelers.

Minor comments:

L73: please replace (McGlynn et al., 2021) by McGlynn et al. (2021)

L81: please replace "at Chan et al. (2011)" by "in Chan et al. (2011)".

L90: please replace "vegetation" by "vegetation type j"

L159: Please make clear what is AMDAR – dataset of observed boundary layer heights?

L162: please describe which airports are IAD and RDU

L194: (4) should be (Figure 4)?

L195: the sentence 'Although there are no isoprene emissions …' does not make sense to me. There are no isoprene emissions shown in Figure 3. Furthermore, Figure 4 actually shows the opposite, i.e. almost zero night-time isoprene concentrations in observations. Should 'observed' be replaced by 'modeled'?

L214: The following statement applies to limonene only, or not? "with a peak in light dependence during the summer and less light-dependence during the rest of the year".

The description of results on Figure 6 is not very clear. Did you interpolate (with 0.01 value step) the modeled emissions or concentration values? Do I understand correctly, that for each month you calculated correlation between observed and modeled values and for a particular month you selected the LDF value that has the highest correlation (and this correlation value is the one shown in the plot)? If yes, please explain better in the text.

L224: "3 monoterpenes: isoprene, α-pinene, β- 225 pinene, and limonene." Please remove isoprene.

p13: Caption of Fig 8 – please review the last sentence.

Supp. material:

- Fig S4 does not show results for January. Please edit the caption.
- Please review the caption of Figure S6. The last sentence does not make sense to me.
- Caption of Table S5 – emphi should be specific font of i?
- Caption of Fig S6 – please review the last sentence.

---

## Author Comment (AC1)

We thank the reviewers for their time and effort in evaluating our research and provide responses to comments and questions below. Reviewer comments are in black and our responses in blue, with new text shown in *italics*. Updated line numbers are noted for each comment.

**RC1**:

Review of "Constraining light dependency in modeled emissions through comparison to observed BVOC concentrations in a southeastern US forest" by Panji et al

Panji et al present a study of methods to improve agreement between modeled and observed biogenic VOC concentrations. The measurements are 1 year's data from Virginia Forest Research Laboratory focussed on isoprene, a-pinene, limonene and sabinene. A chemical box model is used to predict BVOC mixing ratios, using MEGAN to provide the biogenic emissions. The authors find that using MEGAN in default mode underpredicts monoterpene mixing ratios. Interestingly, their observations show that limonene has a different diurnal pattern than expected in summer. The expected diurnal cycle is exhibited by their a-pinene mixing ratio measurements. These measurements peak at night because of the fast OH chemistry during the daytime, reacts a-pinene into other products. Panji et al use the light dependence functions to improve the modeled to observed comparisons, finding that the function varies by compound and with the season.

I like how the study is progressing a one-size-fits all adjustment of the light dependence functions towards species and seasonally varying adjustments. I think the manuscript fits the ACP journal scope. I have one query and a couple of clarification comments before acceptance for publication.

My main confusion is about how the competing actions of limonene emissions and OH chemistry during the day act to keep limonene in (or out of) the air. My understanding is that the reaction of limonene with OH is faster than a-pinene with OH. Is the conclusion here that there is no OH left to quench the daytime limonene at this site, and can this be re-produced in the model?

We would like to thank the reviewer for their comment and have tried to clarify some of these points in the revised manuscript. Reaction rates of both compounds have been measured and are somewhat similar, with limonene reacting roughly twice as fast. We attribute the different diurnal patterns of limonene and α-pinene concentrations during summer months not to depletion of the OH radical, but rather to differences in the light-dependency of the compounds. While both compounds are lost by reaction with OH during the day, we propose that limonene has a stronger relative daytime emission rate compared to α-pinene, as represented by the very different light dependent factors we propose for each compound (nearly 1 vs. 0 for limonene and α-pinene respectively). Due to temperature-dependency of both light-dependent and light-independent emissions, both compounds have strong daytime emissions, but α-pinene also has nighttime emissions and peaks at night because oxidant concentrations are lower, and it accumulates in the shallower boundary layer. Conversely, we suggest that limonene has largely light-dependent emissions during the summer and fall months which implies that it is emitted at higher rates during the daytime and hence presents as having daytime peaks in concentrations despite reacting with its

oxidants. We do not intend to imply that OH is quenched during the day, and do not see such an effect in the model.

L206-209: "Importantly, it is clear that the LDF that best captures observed variability may vary throughout the year, indicating important seasonality. *Though both compounds are reacting with oxidants during the day, the stronger light dependence of limonene yields a high daytime and low nighttime source that produces a daytime peak, while α-pinene has light-independent emissions at night that accumulate in the lower nighttime boundary layer.*"

line 172. I think the statement should read 'independent' instead of 'dependent'? Otherwise it's a strange statement. If they're highly light dependent then there will be no emissions during the night. If they're entirely light independent then the main driver is temperature which usually peaks during the day.

We thank the reviewer for their observation. We have corrected the line to:

L176: "Emissions peak~s~ during the day, even with relatively high fractions of light-*in*dependent emissions (Figure 2)."

Line 195. Figure 4 is quite large but doesn't really get talked about. Unless the new paragraph starting at line 196 is about fig 4? It isn't clear. The new section at line 205 jumps to figure 5.

Line 196 talks about Figure 4 and was labeled incompletely. We have changed it to read as:

L196-198: "Modeled and observed concentrations in the summer months are comparable in magnitude for all three compounds in the summer, and some day-to-day variability is also captured (*Figure* 4)."

Line 215. It's worth mentioning that the correlation coefficient for limonene nearly reaches 1.0 between june to august.

To clarify, the correlation coefficient between modeled and observed limonene concentrations does not reach 1.0 between June to August of 2020, but rather reaches a peak of around 0.5. The solid blue line in Figure 6 represents the Light-Dependent Factor (primary y-axis) which reaches a value of nearly 1. The correlation coefficient is represented by the dashed blue line (secondary y-axis) and reaches a value of roughly 0.5. We have added arrows to the lines for easier interpretation of the figure. We have also highlighted in the revised manuscript that there are many other potential sources of uncertainty that may be impacting the model-measurement agreement, including uncertainty in ozone and $NO_x$ measurements, inherent uncertainty in species-specific emission factors, and uncertainty in planetary boundary layer height. The relative importance, and ability to address these different sources of uncertainty are being explored in upcoming work.

L220-223: "The light dependency of limonene that best correlates with observations varies throughout the year (*solid blue line in* Figure 6*a*), with a peak in light dependence during the summer and less light-dependence during the rest of the year. Conversely, a constant LDF

α-pinene of 0 (i.e., light-independent) throughout the year slightly improves the correlation coefficient *(solid blue line in Figure 6b)."*

L273-280: "Lastly, we note that the simulations fail to capture observed concentrations in the winter months where they are consistently underpredicted. *Lack of regional $NO_X$ and $O_3$ concentrations, uncertainty in the boundary layer height, and poorly constrained temperature-dependent coefficients and emission factors could be reasons for the discrepancies between the modeled and observed concentrations. Furthermore, intra-annual variability of emission factors (Helmig et al., 2013) and other uncertainty in emissions factors due to a scarcity of temperature- and light-dependent emissions measurements for many of these species could explain seasonal disagreements.*  Largely, we are able to capture the day-to-day variability in the concentrations during summer months using this relatively simple setup of MEGAN and a 0-D box model with 7-day rolling average of the boundary layer height conditions, and $NO_X$ and $O_3$ concentrations from non-local sources."

[Figure]

Figure R1/Revised Figure 6. Monthly variation of LDF for limonene and α-pinene estimated by maximizing the correlation between observed concentrations and those estimated by MEGANv3.2 and F0AM at VFRL. The blue circles and red triangles with solid lines represent the time-dependent and default LDF values respectively and the dashed lines represent the corresponding Pearson correlation coefficients *(axes indicated by black arrows with the same line styles).*

Figure 6. Please label the plots with a) and b) as in fig 5.

We thank the reviewer for noting the missing labels. We have added them as shown in Figure R1 (revised Figure 6).

**RC2:**

The submitted manuscript from Panji et al. presents comparison of modeled and measured concentrations of BVOC species, with special focus on monoterpenes, at the Virginia Forest Research Laboratory site (VFRL). The study focuses on the dependence of BVOC emissions on light and therefore proper setting of the light dependence factor (LDF) of individual BVOC compounds in the emission model. To be able to compare the measured concentrations with modeled values, the 0-dimension box model was applied. The authors compare performance of the emission model (MEGANv3.2) through the box model concentrations with MEGAN default LDF values, LDFs obtained from measurements at the VFRL site and LDF values best correlating with the observed concentrations. The paper investigates annual as well as time-varying (monthly) LDF values. The paper suggests new LDF values for selected monoterpene species.

The paper is well written and comprehensively structured. It falls well within the scope of ACP. I suggest accepting the paper for publication after addressing the following questions and minor comments.

Missing measurements of NOx and O3 at the VFRL site at the time of BVOC sampling were substituted by either measurement from previous years or by measurements from 15-53 miles distant stations. Though I understand the need to deal with lack of data, I think these assumptions need at least more discussion of the impact on results. The ozone stations are relatively far away from VFRL with cities such as Charlottesville or Richmond close by that can impact the O3 levels. Do the authors have some evidence that NOx and O3 levels do not have much inter-annual variability at these locations? The NOx and O3 data are crucial for calculation of BVOC concentrations from emissions, therefore can have a substantial impact on the final model to observation comparison.

The reviewer raises an important question. We note that there could be several reasons for discrepancies between modeled and observed concentrations (poorly constrained $NO_X$ or $O_3$ concentrations, dilution, temperature-dependent coefficients, emission factors, etc.) and this has been addressed in Lines 274-278. Some of these sources of uncertainty, such as uncertainty in species-specific emissions factors cannot be easily evaluated. However, we have taken this opportunity to evaluate the impact of non-local measurements as raised by the reviewer. In particular, we examine differences in ozone between the tower and local monitoring sites. Though ozone data coinciding with the model period was not available at the time of this publication, ozone measurements from 2019 are available at multiple tower heights and are compared in Figure R2 to the monitoring site used in the original manuscript. Day-to-day and diurnal variability are closely mirrored between the monitoring site and the tower measurements, with the monitoring site generally reporting 10-20 ppb lower. As most of the analyses in this work focus on temporal variability, the lack of difference in temporal variability between the sites suggests the source of ozone data is not significantly impacting the conclusions of this work. The higher concentrations at the tower, relative to the

monitoring site, suggests that chemical loss is stronger than in the current model, resulting in lower concentrations, as observed for isoprene in a one-week test run shown in Figure R3. Consequently, using ozone from the monitoring station may be impacting model bias, but would not be expected to significantly impact light dependent factors, for which quantification relies on temporal variability (i.e., optimized for correlation coefficients) and is independent of model bias. Apart from that, since ozone levels are constrained by observed measurements, $NO_x$ concentrations have no impact on the simulated VOC levels.

These figures have been included in the supplementary information, as well as discussion of them in the main revised text.

L146-150: *The diurnal patterns at the Albemarle High School station were similar to those measured at the tower during an overlapping period in 2019, with a bias of 10-20 ppb lower at the EPA station relative to that at the tower (Figure S9). Because optimum light dependent factors in this work are determined primarily through correlation and temporal patterns, the similarity in ozone variability between sites does not strongly impact the results of this work, though it does impact model biases (Figure S10)."*

[Figure]

Figure R2/Revised Figure S9. Ozone measured at the Virginia Forest Research Laboratory (VFRL) at different heights from the ground (1, 9, 20, 30, 40 m) and at the EPA monitoring station in Albemarle, VA. (a) average diurnal for all data June through November of 2019, and (b) Sample five-day period showing similar day-to-day variability

[Figure]

Figure R3/Revised Figure S10. Simulated concentrations of isoprene (in ppb) over 5 days in July, 2020 with the ozone used in our model (Baseline $O_3$ denoted by a black line) and at elevated ozone levels (Baseline + 10ppb $O_3$ and Baseline + 20ppb $O_3$ denoted by red and blue lines respectively).

Apart from LDF values and their seasonal changes, there are other parameters of the emission model that can (partially) explain the discrepancy of the modeled and measured results throughout the year. E.g. emission factors. Though not often used that way, EF can also very during the annual cycle (Helmig et al., 2013; https://doi.org/10.1016/j.chemosphere.2013.04.058). The EF intra-annual changes could be another factor that explains a different BVOC concentrations during winter and summer months. Could the authors include this in the discussion?

We thank the reviewer for pointing this out. We agree that intra-annual variation of emission factors could be a reason for seasonal discrepancies in modeled and observed concentrations. Poorly constrained emission factors are an important source of disagreement and have elaborated on that as follows:

L273-280: "Lastly, we note that the simulations fail to capture observed concentrations in the winter months where they are consistently underpredicted. *Lack of regional $NO_X$ and $O_3$ concentrations, uncertainty in the boundary layer height, and poorly constrained temperature-dependent coefficients and emission factors could be reasons for the discrepancies between the modeled and observed concentrations. Furthermore, intra-annual variability of emission factors (Helmig et al., 2013) and other uncertainty in emissions factors due to a scarcity of temperature- and light-dependent emissions measurements for many of these species could explain seasonal disagreements.*  Largely, we are able to capture the day-to-day variability in the concentrations during summer months using this relatively simple setup of MEGAN and a 0-D box model with 7-day rolling average of the boundary layer height conditions, and $NO_X$ and $O_3$ concentrations from non-local sources."

Can the authors please share their opinion (and include it in the paper 'Discussion' or 'Conclusion' section) on why the modeled concentrations are "consistently underpredicted

in the winter months"? Actually, the modeled concentrations of limonene and sabinene are underpredicted also in July (Figs. 7 and 9).

We appreciate the reviewer's insight and agree that this topic deserves further attention. Currently, we do not have a precise understanding of the emission factors, which could be contributing to the underpredicted concentrations. These discrepancies might stem from various sources, including inaccurate standard emission factors in emission inventories, changes in the composition of plants in the forest, or incorrect assumptions about temperature and light dependencies of plant emissions in emission factor models. Light- and temperature-dependent measurements of emissions from many types of vegetation are relatively sparse, and many isomers of monoterpenes or other terpenoids may not be routinely measured. Limonene is generally a relatively well-studied isomer, but it is clear from this study that significant uncertainty remains in its emissions mechanisms and processes. We suspect the bias noted by the reviewer is generally due to these types of uncertainty in species-specific emissions factors. Notably, the most well-studied compounds, isoprene and α-pinene, exhibit more moderate biases, qualitatively supporting the conclusion that increased study of other isomers and their emissions from different vegetation under different conditions may improve agreement. Addressing this issue would benefit from additional flux measurements at the research site, it is currently beyond the scope of this paper. However, this matter presents an opportunity for future studies to provide more accurate representations. This has been included in the Conclusion section of the manuscript as noted in the previous comment.

L227-231: "However, the magnitude of the modeled concentrations remains low, suggesting underpredicted emission rates *and highlights the significant uncertainty in the understanding of its emissions mechanisms and processes. Notably, the most well-studied compounds, isoprene and α-pinene, exhibit more moderate biases, qualitatively supporting the conclusion that increased study of emissions fluxes from different vegetation under different conditions may improve agreement.*"

The authors point out well that the LDF values play an important role in the BVOC models and their precise setting is important in order to obtain sensible emission results. Can the authors please elaborate if the LDF values obtained from the measurements at VFRL can be upscaled from this local site to global representation? If the authors think their values could be used for other studies as well, it would be extremely useful if they could add a Table summarising LDF values per species and per month that they recommend to use according to their study. This would be a very good benefit for other emission modelers.

We thank the reviewer for their valuable suggestions and for highlighting the value of this research. The general conclusion that monoterpene isomers exhibit different light dependence than is the default in these models, and light dependence is seasonally (or

otherwise temporally) variable is likely reasonable to scale up. This is based in part on prior literature of light-dependent emissions from plant species native to other ecosystems, and the highly varied biological functions of terpenoids that would be expected to vary over time. However, we generally believe it would be highly uncertain to scale these quantitative LDF values obtained here to a global representation. It is likely that these values are related to plant speciation and phenology, which may not be comparable in other environments, though application to the southeastern U.S. may be reasonable. A more process-based understanding of light- and temperature-dependent emissions would provide confidence in how to scale these data - in particular, data capturing differences across diverse forest compositions. We would advise caution in applying these directly to global-scale models without referencing locally observed diurnal patterns, but more broadly encourage a more nuanced and variable approach to light dependence.

We have added the requested table in the SI as shown below (Table R1) but emphasize exercising caution about using them directly on other ecosystems.

L250-255: "...the year (as opposed to the 0.2 LDF assumed in the MEGAN model). *The seasonal variation of the compounds examined in this study are presented in Table S8. The general conclusion that monoterpene isomers exhibit different light dependence than is the default in emission models, and that light dependence is seasonally (or otherwise temporally) variable may be applied more broadly or globally. However, we would advise caution in quantitatively applying the values reported here directly to global-scale models without referencing locally observed diurnal patterns as they are likely ecosystem dependent.*"

| Compound | Winter (DJF) | Spring (MAM) | Summer (JJA) | Fall (SON) |
|---|---|---|---|---|
| α-pinene | 0 | 0 | 0 | 0 |
| limonene | 0 | 0.23 | 0.97 | 0.76 |
| β-phellandrene | 0.42 | 0 | 0.44 | 0.71 |
| camphene | 0 | 0 | 0 | 0 |
| tricyclene | 0 | 0.33 | 0 | 0 |
| β-thujene | 0.78 | 0.31 | 0.97 | 1 |
| α-fenchene | 0 | 0 | 0 | 0 |
| sabinene | 0 | 0.24 | 1 | 1 |
| γ-terpinene | | 0.15 | 0.90 | 0.93 |

Table R1/revised Table S8: Seasonal variation of Light-dependent Factors for α-pinene, limonene, β-phellandrene, camphene, tricyclene, β-thujene, α-fenchene, sabinene, and γ-terpinene at VFRL.

Minor comments:

L73: please replace (McGlynn et al., 2021) by McGlynn et al. (2021)

L73: "Further details about the instrumental setup, its calibration and operation are available in  *McGlynn et al. (2021)* and McGlynn et al. (2023b).

L81: please replace "at Chan et al. (2011)" by "in Chan et al. (2011)".

L81: "Further, ecological information of the surrounding forest such as species composition and abundance is available  *in* Chan et al. (2011)."

L90: please replace "vegetation" by "vegetation type j"

L89: "$\chi_j$ represents the fractional area of a model grid cell covered with vegetation *type j*."

L159: Please make clear what is AMDAR – dataset of observed boundary layer heights?

L162-164: "*Boundary layer height* observations in the AMDAR dataset are available regularly but non-continuously throughout the year, and estimation of boundary layer height requires non-trivial data analyses."

L162: please describe which airports are IAD and RDU

L165-167: "An average boundary layer height at  *the Washington Dulles International Airport (*IAD*, near Washington, D.C.)* and *the Raleigh-Durham International Airport (*RDU*, near Raleigh, NC)* are used."

L194: (4) should be (Figure 4)?

L196-198: "Modeled and observed concentrations in the summer months are comparable in magnitude for all three compounds in the summer, and some day-to-day variability is also captured (*Figure* 4)."

L195: the sentence 'Although there are no isoprene emissions …' does not make sense to me. There are no isoprene emissions shown in Figure 3. Furthermore, Figure 4 actually shows the opposite, i.e. almost zero night-time isoprene concentrations in observations. Should 'observed' be replaced by 'modeled'?

We thank the reviewer for their comment. We note that the text referenced the wrong figures and using 'modeled' avoids confusion. It has been corrected as follows:

L200-201: "Although there are no isoprene emissions during night-time (Figure  *2*), non-zero night-time isoprene concentrations are  *modeled*, which may suggest that nighttime chemistry and/or dilution may not be fully captured (Figure  *3*)."

L214: The following statement applies to limonene only, or not? "with a peak in light dependence during the summer and less light-dependence during the rest of the year".

We thank the reviewer for seeking clarification. In this discussion, the statement "with a peak in light dependence during the summer and less light-dependence during the rest of the year" applies to limonene. We explore the seasonal variability of LDFs for different compounds in section 3.4. We have clarified as follows:

L220: "The light dependency *of limonene* that best correlates with observations varies throughout the year…"

The description of results on Figure 6 is not very clear. Did you interpolate (with 0.01 value step) the modeled emissions or concentration values? Do I understand correctly, that for each month you calculated correlation between observed and modeled values and for a particular month you selected the LDF value that has the highest correlation (and this correlation value is the one shown in the plot)? If yes, please explain better in the text.

Yes, that is the correct interpretation of our work. We have added clarifications regarding the dashed and the solid lines in Figure 6 by adding arrows to the figure to denote the y-axis they represent and further explained it in the caption. Please refer to Figure R1 of this document for the changes made.

L216-220: "To better quantify the monthly variation of LDF with the highest correlation with observed concentrations,  *the modeled concentrations at the six LDF values (ranging from 0 to 1) described before were linearly interpolated in 0.01 increments to achieve a higher resolution. These interpolated concentrations were then compared with observed concentrations to find the LDF that produced the highest correlation.*"

L224: "3 monoterpenes: isoprene, α-pinene, β- 225 pinene, and limonene." Please remove isoprene.

L235: "Currently, MCM3.3.1 used during F0AM simulations contains only isoprene and 3 monoterpenes:  α-pinene, β-pinene, and limonene."

p13: Caption of Fig 8 – please review the last sentence.

"The blue and red *solid* lines present the LDF values and the dashed lines represent the corresponding Pearson correlation coefficients."

Supp. material:

- Fig S4 does not show results for January. Please edit the caption.

    "A snapshot of α-pinene concentrations (in ppb) for  July and September of 2020 using the monthly LDF as shown in Figure 3.6. The Pearson correlation coefficient values of the adjusted and default modeled (refer to Table 3.1)

concentrations against the observed concentrations are reported as $R_{AdjustedLDF}$ and $R_{DefaultLDF}$."

- Please review the caption of Figure S6. The last sentence does not make sense to me.

"The blue and red *solid* lines present the LDF values and the dashed lines represent the corresponding Pearson correlation coefficients"

- Caption of Table S5 – emphi should be specific font of i?

The 'i' denoting compound in Table S5 has been emphasized.

- Caption of Fig S6 – please review the last sentence.

"The blue and red *solid* lines present the LDF values and the dashed lines represent the corresponding Pearson correlation coefficients"